# Non-allelic gene conversion enables rapid evolutionary change at multiple regulatory sites encoded by transposable elements

**Christopher E Ellison, Doris Bachtrog\***

Department of Integrative Biology, University of California, Berkeley, Berkeley, United States

**Abstract** Transposable elements (TEs) allow rewiring of regulatory networks, and the recent amplification of the ISX element dispersed 77 functional but suboptimal binding sites for the dosage compensation complex to a newly formed X chromosome in Drosophila. Here we identify two linked refining mutations within ISX that interact epistatically to increase binding affinity to the dosage compensation complex. Selection has increased the frequency of this derived haplotype in the population, which is fixed at 30% of ISX insertions and polymorphic among another 41%. Sharing of this haplotype indicates that high levels of gene conversion among ISX elements allow them to 'crowd-source' refining mutations, and a refining mutation that occurs at any single ISX element can spread in two dimensions: horizontally across insertion sites by non-allelic gene conversion, and vertically through the population by natural selection. These results describe a novel route by which fully functional regulatory elements can arise rapidly from TEs and implicate non-allelic gene conversion as having an important role in accelerating the evolutionary fine-tuning of regulatory networks.

**\*For correspondence:**
dbachtrog@berkeley.edu

**Competing interests:** The authors declare that no competing interests exist.

## Introduction

A substantial portion of animal genomes is composed of repetitive sequences, including gene duplicates, satellite DNA, and transposable elements. Gene conversion is a major force shaping the evolution of repetitive regions, and interlocus or non-allelic gene conversion between sequence duplicates has been studied extensively for its role in concerted evolution (*Chen et al., 2007*; *Ohta, 2010*). Non-allelic gene conversion also affects selection operating in gene families. Compared to single-copy genes, a family of gene duplicates presents a larger mutational target, and a mutation arising in any gene copy can be spread among copies by non-allelic gene conversion, thereby increasing the efficiency of both positive and purifying selection (*Mano and Innan, 2008*). Non-allelic gene conversion homogenizes the arrays of ribosomal DNA gene copies present in the genomes of most organisms (*Eickbush and Eickbush, 2007*), has generated allelic diversity within the human leukocyte antigen gene family (*Zangenberg et al., 1995*), and has allowed palindromic genes on the human Y chromosome to escape degeneration (*Rozen et al., 2003*).

Transposable elements give rise to families of duplicate sequences. A propensity for some TEs to carry regulatory motifs and to insert adjacent to coding sequence gives them the potential for being potent modulators of gene regulatory networks (*Feschotte, 2008*; *Cowley and Oakey, 2013*). The regulatory elements provided by these TEs, however, may be suboptimal in function, and subject to subsequent fine-tuning (*Polavarapu et al., 2008*). Unlike regulatory elements where short binding motifs (10 basepairs on average for transcription factors; *Stewart et al., 2012*) evolve de novo via point mutation or microsatellite expansion, binding sites that evolve from TEs are initially almost identical in sequence and are nested within a larger repeat unit (hundreds or thousands of basepairs in size), and may thus be subject to non-allelic gene conversion. Re-wiring of the dosage compensation

**eLife digest** Mutations change genes and provide the raw material for evolution. Genes are sections of DNA that contain the instructions for making proteins or other molecules, and so determine the physical characteristics of each organism. Genetic mutations that increase an organism's number of offspring and chances of survival are more likely to be passed on to future generations. Changes to when or where a gene is switched on (so-called regulatory mutations) can also provide fitness benefits and can therefore be selected for during evolution.

Transposable elements are sequences of DNA that are also called 'jumping genes' because they can make copies of themselves and these copies of the transposable element can move to other locations in the genome. Some transposable elements contain sequences that switch on nearby genes. If different copies of a transposable element that contains such a regulatory sequence insert themselves in more than one place, it can result in a network of genes that can all be controlled in the same way. The regulatory sequences contained within transposable elements are not always optimal, but they can be fine-tuned through evolution.

A fruit fly called *Drosophila miranda* has a transposable element called ISX that has, over time, placed up to 77 regulatory sequences around one of this species' sex chromosomes. Just as in humans, female flies are XX and males are XY; but having only one copy of the X chromosome means that male flies need to increase the expression of certain genes to produce a full-dose of the molecules made by the genes. This process is called dosage compensation and in 2013 the 77 ISX regulatory sequences on the fruit fly's X chromosome were shown to help recruit the molecular machinery that carries out dosage compensation to nearby genes, albeit inefficiently. Now Ellison and Bachtrog—who also conducted the 2013 study—report how these transposable elements have been fine-tuned to make them more effective for dosage compensation.

Ellison and Bachtrog uncovered two mutations that make the ISX transposable element better at recruiting the dosage compensation molecular machinery. ISX spread around different locations along the fly's X chromosome before these mutations arose; this means that initially none of the 77 insertions carried the two mutations, but now 30% of the 77 elements have the mutations in all flies, and 41% have them in only some flies.

The same mutations have spread between the different ISX elements because transposable elements with the mutations have been used to directly convert other ISX elements without them. These mutations have also become more common in the fruit fly population by being passed on to offspring and increasing their survival. These two routes have accelerated the fine-tuning of these transposable elements for use in gene regulation. This implies that regulatory sequences derived from transposable elements evolve in a way that is fundamentally different from those that arise by other means, as the direct conversion between these insertions allows fine-tuning mutations to spread more rapidly.

network in *Drosophila miranda* was driven by TE-mediated amplification of a functional but suboptimal binding motif (*Ellison and Bachtrog, 2013*). Here we show that non-allelic gene conversion is catalyzing the rapid fine-tuning of these suboptimal motifs by allowing sequence variants that optimize binding affinity to spread among elements.

Dosage compensation in Drosophila is mediated by a male-specific ribonucleoprotein complex (the male-specific lethal or MSL complex) that binds to a GA-rich sequence motif (the MSL recognition motif) at a number of chromatin entry sites on the X chromosome (*Alekseyenko et al., 2008*; *Straub et al., 2008*). We previously studied the acquisition of novel chromatin entry sites on newly formed X chromosomes in *D. miranda*, a species where two independent sex chromosome/autosome fusions resulted in a karyotype composed of three X chromosome arms, each of a different age (*Alekseyenko et al., 2013*; *Zhou et al., 2013*). XL is homologous to the X chromosome of *Drosophila melanogaster* and has been a sex chromosome for at least 60 million years (*Richards et al., 2005*); chromosome XR formed roughly 15 million years ago when an autosome (Muller element D) fused to XL (*Carvalho and Clark, 2005*), and the neo-X/neo-Y chromosome pair originated around 1.5 million years ago when the Y fused to another autosome (Muller element C) (*Bachtrog and Charlesworth, 2002*). Dosage compensation evolved on both XR and the neo-X shortly after their emergence, through acquisition of

novel chromatin entry sites and co-option of the MSL regulatory network (*Bone and Kuroda, 1996*; *Marin et al., 1996*). Interestingly, we discovered that the acquisition of dosage compensation on both XR and the neo-X chromosome was in part mediated by the independent domestication of helitron transposable elements that contained MSL recognition motifs, which we have termed ISXR and ISX, respectively (*Ellison and Bachtrog, 2013*).

ISX is highly enriched on the neo-X chromosome of *D. miranda* and is derived from the abundant ISY element. Compared to ISY, ISX contains a 10 basepair deletion that creates a MSL recognition motif, thereby allowing it to act as a chromatin entry site (*Ellison and Bachtrog, 2013*). Our previous study showed that while amplification of ISX about 1 million years ago provided dozens of functional chromatin entry sites on the neo-X chromosome of *D. miranda*, the motif dispersed by ISX is distinct from the canonical motif that is enriched within chromatin entry sites on XL and XR, and shows significantly lower affinity to the MSL complex compared to motifs on XL and XR (*Ellison and Bachtrog, 2013*). For these reasons, we postulated that the ISX binding motif is suboptimal, and predicted that refining mutations should accumulate within each MSL recognition motif until the neo-X chromosome becomes fully dosage compensated (*Ellison and Bachtrog, 2013*).

## Results

### Variation at MSL recognition motifs among ISX insertions in *D. miranda* strain MSH22

To identify potential refining mutations that optimize MSL-binding at chromatin entry sites derived from the ISX element, we characterized sequence variation within the MSL recognition motifs and flanking sequence regions for all 77 insertions of the ISX element on the neo-X chromosome in the sequenced reference strain MSH22 (*Figure 1A*). Because we have previously demonstrated that ISX contains a functional MSL recognition motif but the closely related ISY element does not (*Ellison and Bachtrog, 2013*), we sought to identify sequence variants that were present in multiple ISX elements but rare or absent in ISY elements from the same chromosome.

Using these criteria, we identified a sequence haplotype adjacent to the MSL recognition motif that is common among MSH22 ISX insertions and rare among ISY elements: 57% of ISX elements carry this haplotype vs 0.7% of neo-X ISY insertions, an asymmetry significantly different from that expected by chance (Fisher's Exact Test; p < 2.2e-16). The haplotype consists of two mutations (G → T and A → T), separated by two basepairs, which are in perfect linkage disequilibrium among ISX but not ISY elements (*Figure 1* and *Figure 1—figure supplement 1*). Because ISX is descended from ISY and the TT alleles are rare among ISY elements, they are likely to be derived. We hereafter refer to these mutations as the TT haplotype.

### The TT haplotype increases MSL complex binding affinity

To determine if the TT haplotype affects binding affinity of the MSL complex, we used published ChIP-seq data of MSL3 (a component of the MSL complex) from *D. miranda* strain MSH22 (*Alekseyenko et al., 2013*). We compared in vivo MSL complex binding levels for the 44 MSH22 ISX insertions carrying the TT haplotype to the 33 insertions with the ancestral GA haplotype. The insertions with the TT alleles had significantly higher levels of MSL complex binding compared to those with the GA alleles (Wilcoxon test p = 0.01; *Figure 2A*).

We previously demonstrated that insertion of an ISX element in the *D. melanogaster* genome results in recruitment of the MSL complex to an ectopic autosomal location (*Ellison and Bachtrog, 2013*). We used this same system to dissect the relationship between the TT alleles and MSL complex binding affinity. Starting with a cloned ISX element (*Ellison and Bachtrog, 2013*), we used site-directed mutagenesis to create variants of this element that differ only with respect to the TT haplotype. Each of the four possible haplotypes (GA, GT, TA, and TT) was engineered and inserted onto *D. melanogaster* chromosome 2L at cytosite 38F1 using recombinase mediated cassette exchange (RMCE) (*Bateman et al., 2006*). We then measured the effect of each of the derived variants by quantifying allele-specific binding levels of the MSL complex in F1 hybrids between the ancestral haplotype (GA) and each of the derived haplotypes (GT, TA, and TT).

Interestingly, each T allele, when assayed separately, has a negative effect on MSL binding levels compared to the ancestral G or A allele (*Figure 2B*). However, when combined, the TT haplotype results in significantly increased levels of MSL complex binding, relative to the ancestral GA haplotype

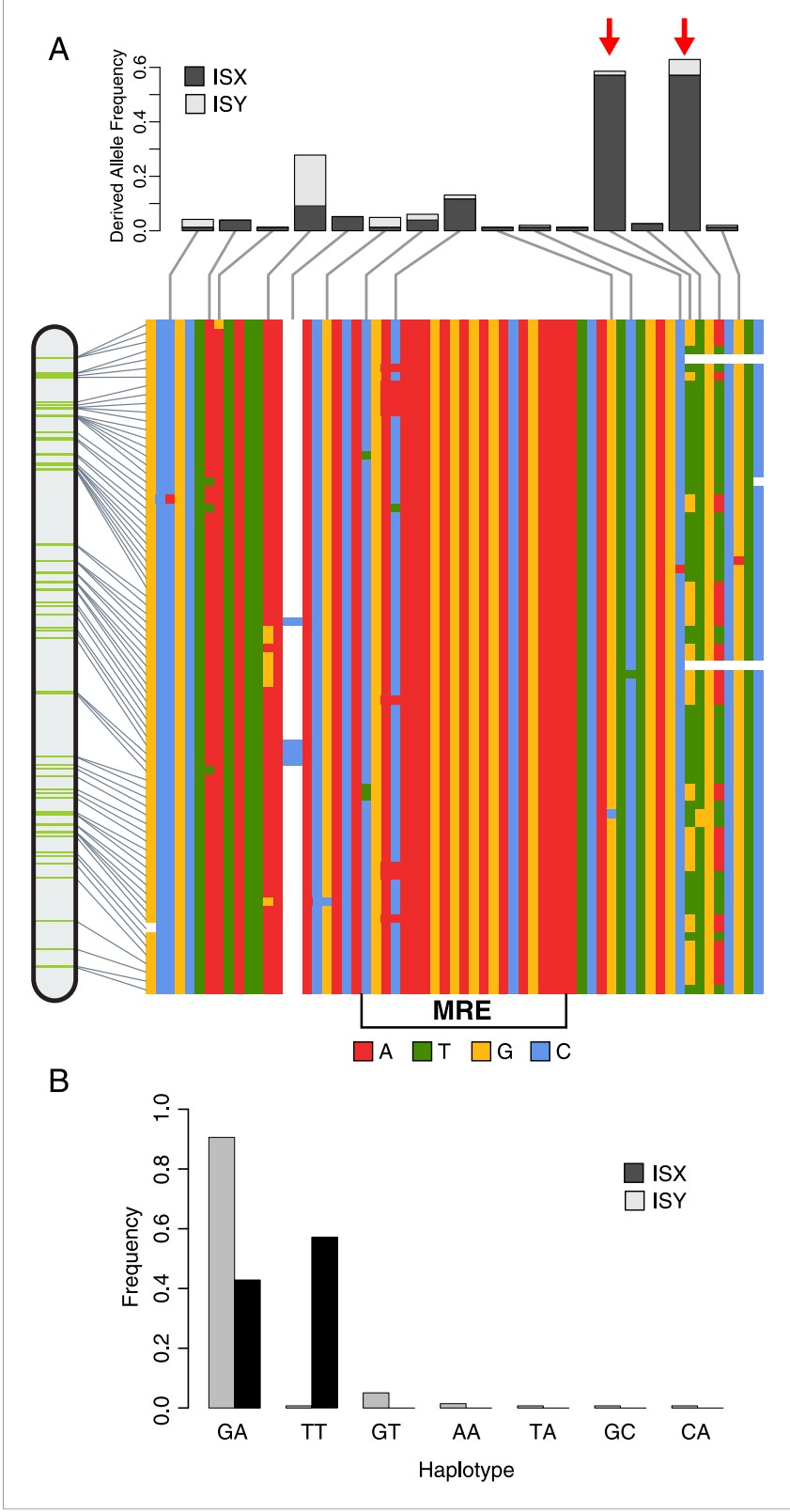

**Figure 1**. TE-derived MSL recognition element (MRE) motifs from the neo-X chromosome of *Drosophila miranda*. (**A**) The MSL recognition motif (MRE) plus 20 basepairs of flanking sequence were extracted from all 77 ISX

*Figure 1. continued on next page*

*Figure 1. Continued*

transposable elements located on the neo-X chromosome in the MSH22 reference genome assembly. The multiple sequence alignment of these 77 sequence regions (arranged from top-to-bottom in the order in which they are found on the chromosome) shows that there is sequence variation among elements both within and adjacent to the 21 basepair MRE motif. Each variant has been classified as ancestral or derived based on its frequency in the ISX progenitor element, ISY. The derived allele frequency for each variant in this region is shown for ISX as well as 139 ISY elements from the neo-X chromosome (see *Figure 1—figure supplement 1* for ISY alignment). Red arrows point to the derived TT haplotype that is common among ISX elements but rare in ISY. (**B**) Barplot showing the frequencies of all haplotypes at the GA/TT sites, for ISY and ISX elements separately. Two haplotypes are present within ISX elements (GA and TT) and the two alleles within each haplotype are in perfect linkage disequilibrium. In contrast, the majority of ISY elements harbor the GA haplotype, but these two alleles are not in perfect linkage disequilibrium among ISY elements. Rather, five additional allelic combinations are present at low frequencies in this location among ISY, but not ISX elements.

The following figure supplement is available for figure 1:

**Figure supplement 1**. Alignment of ISY elements from the *D. miranda* MSH22 genome assembly.

(Wilcoxon Test p = 0.0289; *Figure 2B*). These results suggest that there is sign epistasis between the two alleles and that the high frequency TT haplotype represents a refining/fine-tuning adaptation, since recruitment of MSL complex to the adjacent MSL recognition motif is increased.

## Non-allelic gene conversion is spreading the TT haplotype among ISX insertions

It is unlikely that the TT haplotype arose multiple times by parallel mutation, and there are two possibilities that could explain its prevalence among MSH22 ISX insertions. First, this double mutation may have occurred early during the process of ISX amplification, thus giving rise to two lineages of ISX: one that carries the ancestral GA haplotype, and the other with the TT haplotype. The TT-harboring

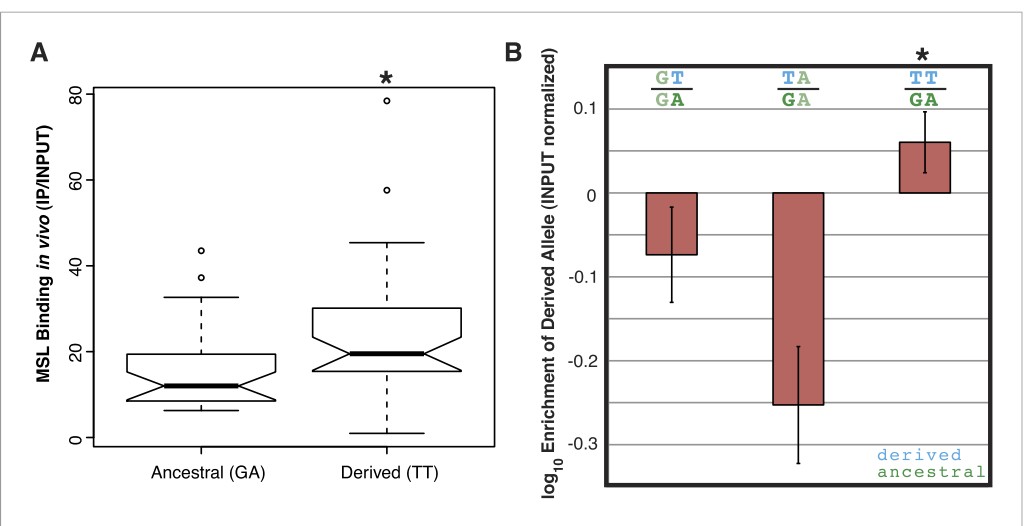

**Figure 2**. The TT haplotype increases MSL binding affinity. (**A**) MSL3 ChIP-seq data from *D. miranda* strain MSH22 shows that the ISX insertions carrying the TT haplotype recruit significantly higher levels of MSL complex compared to those with the GA haplotype (Wilcoxon test p = 0.01). (**B**) Engineered ISX elements that differ only with respect to the TT haplotype bind different levels of MSL complex. There is an epistatic interaction between the two 'T' alleles such that separately, they decrease MSL complex binding relative to the ancestral allele, but together in the TT haplotype, they increase MSL complex binding (Wilcoxon Test p = 0.028 for both comparisons [GT vs TT and TA vs TT]). The rectangles and error bars show the average and standard deviation of values from four biological replicates for each condition.

elements in MSH22 would then all be descendants from the latter ISX lineage. Alternatively, this mutation may have occurred only after the GA-containing ISX element was fixed in the population at all 77 neo-X insertion sites, at which point it was spread among independent ISX elements via non-allelic gene conversion.

We can distinguish between these possibilities by examining patterns of sequence polymorphism for each ISX insertion across multiple strains of *D. miranda*. A canonical signature of non-allelic gene conversion is the presence of shared polymorphisms across sequence duplicates (*Arguello et al., 2006*; *Mansai and Innan, 2010*). If gene conversion is spreading the TT haplotype among ISX insertions, we expect it to be polymorphic among individuals at several ISX insertion sites, whereas we do not expect the TT haplotype to be polymorphic at individual ISX insertions under the alternative scenario.

To genotype multiple wild-derived individuals at each of the MSH22 ISX insertions, we used paired-end Illumina genomic resequencing data from 23 inbred lines of *D. miranda*, including MSH22. We aligned all reads to the MSH22 reference genome and identified mate-pairs where one mate was anchored in unique sequence flanking an ISX insertion. We then assembled these reads to generate a contig spanning the 5′ flank of the ISX element insertion, which contains the MSL recognition motif, for each inbred line. Using this approach we generated population data for 66 insertions out of the 77 total ISX insertions present in the MSH22 reference genome assembly. Uneven sequence coverage between insertions and individuals meant that not all insertions could be assembled for each individual. However, the majority of individuals are represented in the majority of datasets: each insertion dataset contained ~20 lines on average (see Dataset S1 in Dryad: *Ellison and Bachtrog, 2015*). Almost all ISX insertions are fixed among strains (65 of 66) and insertion sites are identical between lines, suggesting that independent parallel insertions are unlikely to be present within our dataset. We performed PCR and Sanger sequencing on a subset of these regions and estimate the base-calling error rate of our Illumina contigs to be ~0.1%.

Consistent with non-allelic gene conversion spreading the TT haplotype, we observe a strong signal of allele-sharing within the sequence region flanking the MSL recognition motif among ISX insertions (*Figure 3*). On average, 68.9% of polymorphisms observed within a given insertion are shared among other insertions (though most polymorphisms are shared only between a few elements). The TT haplotype is especially striking in this regard as it is polymorphic in 41% of insertions (*Figure 3*, *Figure 4* and *Figure 3—figure supplement 1*). If population subdivision contributes to this excess of allele sharing, we would expect individuals to cluster by allele state at the TT locus, across all polymorphic ISX insertions. Instead, we find that different individuals contribute to the TT polymorphism at each of these ISX insertions (*Figure 4—figure supplement 1*), suggesting that abundant non-allelic gene conversion is the most likely explanation for this observation. Interestingly, the population frequency of the TT haplotype is similar among insertions that are near each other on the chromosome (permutation test p = 0.018; *Figure 4*). This is consistent with higher gene conversion rates between more closely linked ISX elements generating correlated population frequencies among adjacent elements (*Sasaki et al., 2010*).

## Selection is driving the spread of the TT haplotype through the population

To test if selection has acted to increase the frequency of the TT haplotype in the population, we examined patterns of polymorphisms at GA- and TT-containing ISX elements. The TT haplotype harbors significantly less linked variation than the ancestral GA haplotype, across insertion sites and individuals (haplotype diversity = 0.53 vs 0.81; resampling p < 0.001; *Figure 5A*). In addition, ISX insertions where TT is fixed have significantly lower nucleotide diversity compared to the insertions where GA is fixed (one-sided Wilcoxon test p = 0.035; *Figure 5B*). Finally, the frequency spectrum at the TT haplotype also shows an excess of high frequency derived alleles, compared to the frequency spectrum at the GA haplotype (resampling p = 0.027; *Figure 5C*). All of these patterns are expected if natural selection acting on the TT haplotype is driving its spread through the population.

## Discussion

Recent work in a variety of eukaryotes suggests that transposable elements may be major drivers of regulatory evolution (*Feschotte, 2008*; *Cowley and Oakey, 2013*). Their high transposition rate and

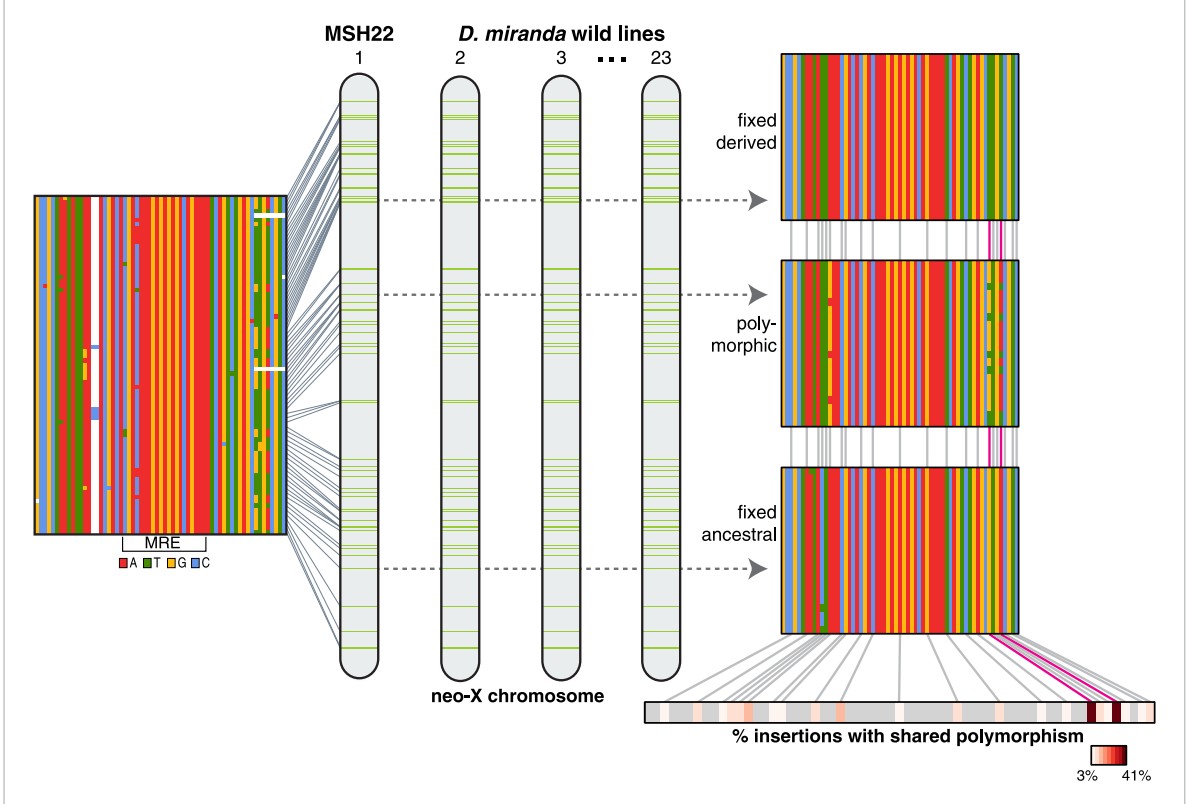

**Figure 3**. ISX variation among wild lines of *D. miranda*. For each ISX insertion identified within the *D. miranda* MSH22 reference genome assembly (alignment shown at left, see also *Figure 1*), we characterized sequence variation across *D. miranda* individuals. The TT haplotype (magenta lines) was fixed across individuals at 30% of insertions (see example alignment, top right), polymorphic at 41% of insertions (example shown middle right), and absent at 29% of insertions (bottom right). Allele sharing between insertions occurs at sites other than the TT haplotype, but these sites tend to be shared across fewer insertions (see heatmap, bottom right). *Figure 3—figure supplement 1* shows the population alignment across all ISX insertions on the neo-X.

The following figure supplement is available for figure 3:

**Figure supplement 1**. Shared polymorphism across sixty-nine ISX insertions.

ability to supply ready-to use regulatory elements across the genome implies that they may rapidly wire new genes into regulatory networks (*Feschotte, 2008*). We recently showed that domesticated TEs contribute to rewiring of the dosage compensation network in *D. miranda*, but appear to supply only suboptimal binding sites for the MSL complex (*Ellison and Bachtrog, 2013*). Here, we identify a derived haplotype with two mutations that interact epistatically to increase binding affinity for the MSL complex. We show that these fine-tuning mutations spread among independent ISX insertions by non-allelic gene conversion, and through the population by natural selection (*Figure 6*). Relative to regulatory elements that evolve in isolation, a family of regulatory motifs dispersed by TEs presents a larger mutational target, and a mutation arising in any element contained within a larger repeat unit (the TE) can spread among copies by non-allelic gene conversion. Consequently, the rate of evolutionary fine-tuning at such regulatory elements can be greatly accelerated by increasing their effective population size (*Mano and Innan, 2008*). Thus, transposable elements can 'crowd-source' beneficial mutations to rapidly fine-tune regulatory networks.

Our transgenic experiments show that each individual T allele actually decreases the binding affinity for the MSL complex relative to the ancestral GA haplotype. Thus, TA or GT haplotypes should be selected against in the population if present on a functional ISX element. Consistent with the deleterious effect of individual T alleles, the TA and GT haplotypes are present on some ISY elements but completely absent from ISX, that is, we find the two T mutations to be in perfect linkage

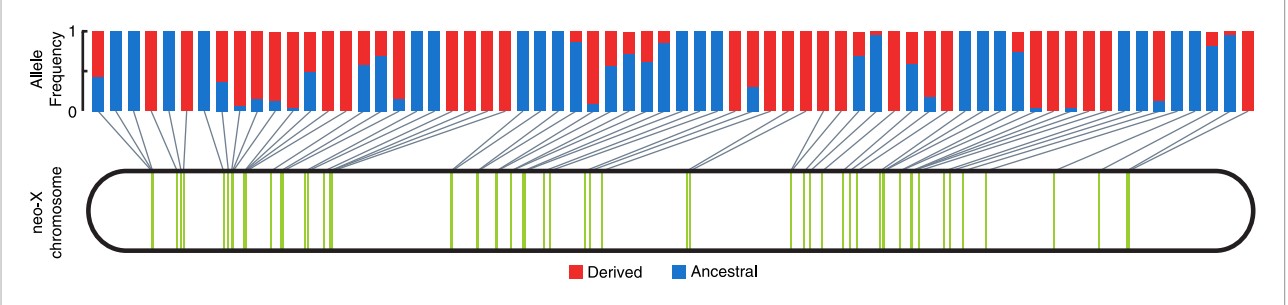

**Figure 4**. Population frequency of TT haplotype across ISX insertions. The location of all ISX elements on the *D. miranda* neo-X chromosome, as inferred from the MSH22 reference genome assembly, is shown by vertical green bars. The derived TT haplotype (frequency shown in red), is polymorphic at 27 of 66 ISX insertions, a pattern consistent with non-allelic gene conversion.

The following figure supplement is available for figure 4:

**Figure supplement 1**. ISX genotype across insertions and individuals.

disequilibrium among ISX elements but not ISY (*Figure 1B*). While most ISY elements carry the ancestral GA haplotype, a small fraction (0.7% of neo-X ISY insertions) instead carry the derived TT haplotype. It is therefore possible that the TT haplotype was introduced onto the ISX background by non-allelic gene conversion from ISY. Under this scenario, the large family of ISY elements in the *D. miranda* genome could be acting as a reservoir of natural variation, where complex mutations can accumulate in the absence of epistasis. Non-allelic gene conversion could then transfer these haplotypes to related repetitive elements (such as ISX). While many of these haplotypes are likely to be neutral or deleterious, some may be beneficial, as in the case of the TT haplotype. Such a scenario avoids the waiting time for a double mutation, as well as the fitness valley that would have to be traversed if the two mutations were to occur sequentially on the ISX background.

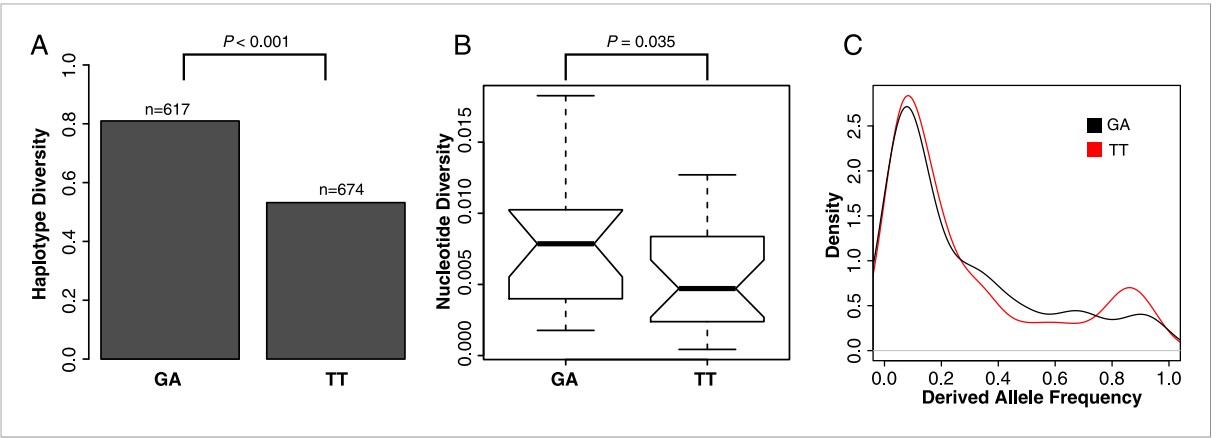

**Figure 5**. Selection shapes patterns of variation at the TT haplotype. (**A**) Haplotype diversity across all ISX sequences. Assembled ISX contigs were combined for all insertions and individuals. The 25 basepairs flanking each side of the TT region were extracted from a total of 1291 sequences and split into two groups based on whether they contained the TT or GA haplotype. Haplotype diversity was then calculated for each group. The difference between groups is significantly larger than expected by chance (resampling p < 0.001), with the sequences containing the TT haplotype having less haplotype diversity compared to those containing the GA haplotype. (**B**) Nucleotide diversity across all ISX sequences. We compared nucleotide diversity for ISX insertions where all individuals carried the ancestral GA haplotype to those where the derived TT haplotype was fixed. ISX insertions that are fixed for the TT haplotype have significantly reduced nucleotide diversity compared to insertions fixed for the GA haplotype (one-sided Wilcoxon test p = 0.035). (**C**) Allele-frequency spectrum across ISX sequences. The allele frequency spectrum was calculated separately for TT and GA-carrying ISX elements, across all insertions and individuals, using the first 200 basepairs of ISX sequence. Consistent with incomplete hitchhiking under positive selection, the TT frequency spectrum shows an excess of high frequency derived alleles, compared to the GA spectrum (resampling p = 0.027).

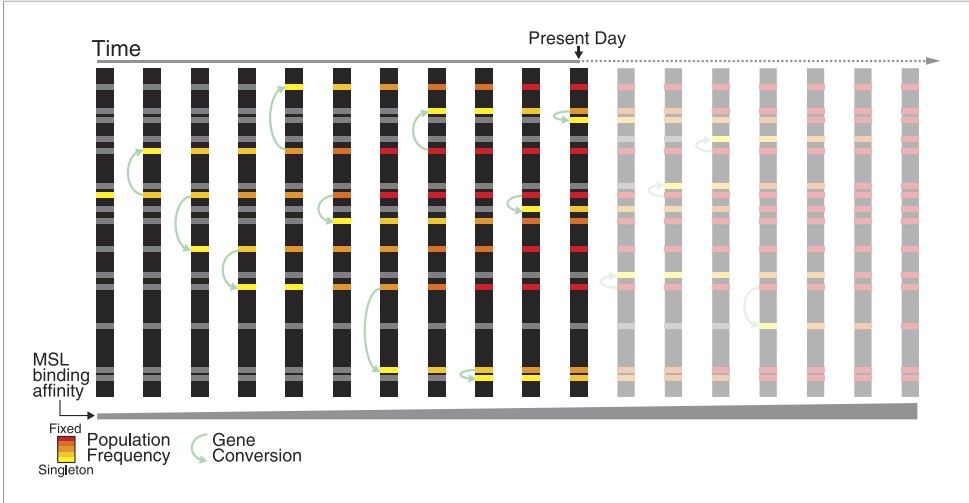

**Figure 6**. Non-allelic gene conversion spreads refining mutations among TE-derived MSL recognition motifs. Shared polymorphism of the TT haplotype among ISX insertions suggests a model where a mutation that refines regulatory activity arose once at a single TE-derived regulatory element, and spread across elements via non-allelic gene conversion. Over evolutionary time, such a mutation spreads in two dimensions: horizontally among TE-derived regulatory elements and vertically through the population, until it is fixed across elements and across individuals. The TT haplotype is at the midpoint of this process. Across ISX insertions, it is fixed, absent, and polymorphic, in approximately equal proportions.

To conclude, our findings suggest that TE-dispersed binding motifs follow an evolutionary trajectory that is fundamentally different from those that arise by other means. The complementary roles of TEs in dispersing regulatory motifs, and gene conversion in spreading subsequent refining mutations, combine to allow for the rapid rewiring and fine-tuning of gene regulatory networks. This process adds a new layer of complexity onto how TEs influence regulatory innovation, as well as a new context in which gene conversion affects genome evolution.

## Materials and methods

### Resequencing of *D. miranda* wild lines

Isofemale lines were established from individuals collected in Northern California and inbred for several generations. DNA was extracted from 1–8 females per line using the Qiagen PureGene kit (Netherlands) and fragmented by nebulization. Paired-end Illumina libraries were constructed using standard protocols (*Bentley et al., 2008*) and sequenced on an Illumina Genome Analyzer II machine (San Diego, CA).

### ISX assembly and variant identification

Resequencing data were mapped to version 2.2 of the *D. miranda* MSH22 reference assembly (GenBank: AJMI00000000.2) using bowtie2 (*Langmead and Salzberg, 2012*). ISX locations were identified in *Ellison and Bachtrog (2013)*. Paired-end read alignments were evaluated within 2 kilobase windows flanking each ISX insertion and reads with mapping quality of 20 or greater were extracted along with their mate. The extracted mate pairs were then assembled using IDBA-UD, for each line separately (*Peng et al., 2012*). Contigs were aligned using FSA (*Bradley et al., 2009*) and visualized with Jalview (*Waterhouse et al., 2009*). A custom Perl script (available at https://github.com/chris-ellison/MSAvariants) was used to identify sequence variants within the alignments. We also PCR amplified eight of the ISX insertions where the TT haplotype was polymorphic. We confirmed that this polymorphism was present at each of these insertions and estimated the base-calling accuracy of the assemblies by sequencing the PCR products using Sanger technology.

## Transgenesis

We used the QuikChange Lightning site-directed mutagenesis kit from Agilent Technologies (Santa Clara, CA) and the ISX element cloned in *Ellison and Bachtrog (2013)* to engineer four ISX variants that differed only with respect to the TT haplotype: ISX-GA, ISX-GT, ISX-TA, and ISX-TT. Each construct was injected by BestGene Inc. (Chino Hills, CA) into *D. melanogaster* embryos carrying a RMCE landing site at cytosite 38F1 on chromosome 2L (Bloomington Drosophila Stock Center strain #27388). Transformants were verified by PCR and Sanger sequencing.

## Quantification of allele-specific binding levels of the MSL complex

Male third instar larvae (~250 mg) were collected from F1 hybrids between ISX-GA and each of the other three engineered lines: ISX-GT, ISX-TA, and ISX-TT. Chromatin immunoprecipitation was performed for four biological replicates of each of these lines using the MSL2 d-300 primary antibody from Santa Cruz Biotechnology Inc. (Santa Cruz, CA) and the protocol described in *Alekseyenko et al. (2013)*. Primers flanking the ISX MRE region were used to generate heterozygous amplicons from the MSL2 IP and input control. Sanger chromatograms were used in conjunction with polySNP software (*Hall and Little, 2007*) to calculate relative abundance of ISX alleles within the IP and input control amplicons. Abundance of the 'T' alleles in the IP amplicons relative to the ancestral G/A alleles was calculated and normalized by the same values from the input control.

## Permutation and resampling tests

To determine if the TT frequency among neighboring ISX elements was correlated, we clustered elements within 100 kb of each other and calculated the standard deviation in TT allele frequency within clusters. We then compared these values to 1000 permutations where TT allele frequency was randomly shuffled between ISX locations.

The haplotype diversity and allele frequency spectrum resampling tests were performed by drawing, without replacement, two groups of size 617 and 674, respectively, from the pool of 1291 ISX sequences. The intergroup difference in haplotype diversity, as well as the number of derived variants with frequency of 0.75 or greater, was calculated for each of 1000 replicates and compared to the difference between the TT and GA groups.

## Acknowledgements

This work was funded by NIH grants (R01GM076007 and R01GM093182) to DB and a NIH postdoctoral fellowship to CEE. All DNA-sequencing reads generated in this study are deposited at the National Center for Biotechnology Information Short Reads Archive (www.ncbi.nlm.nih.gov/sra) under the BioProject ID PRJNA270105. We thank Molly Przeworski, Jeffrey Fawcett, Isabel Gordo and Monty Slatkin for comments on the manuscript and Daniel Weissman for helpful discussions.

## Additional information

### Funding

| Funder | Grant reference | Author |
| --- | --- | --- |
| National Institutes of Health (NIH) | R01GM076007 | Doris Bachtrog |
| National Institutes of Health (NIH) | R01GM093182 | Doris Bachtrog |
| National Institutes of Health (NIH) | 1F32GM103186-01 | Christopher E Ellison |

The funder had no role in study design, data collection and interpretation, or the decision to submit the work for publication.

### Author contributions

CEE, DB, Conception and design, Acquisition of data, Analysis and interpretation of data, Drafting or revising the article

## Additional files

### Major datasets

The following datasets were generated:

| Author(s) | Year | Dataset title | Dataset ID and/or URL | Database, license, and accessibility information |
|---|---|---|---|---|
| Ellison CE, Bachtrog D | 2015 | Data from: Multiple sequence alignment of ISX transposable elements | 10.5061/dryad.dg483 | Available at Dryad Digital Repository under a CC0 Public Domain Dedication. |
| Ellison CE, Bachtrog D | 2015 | DNA-sequencing reads | http://www.ncbi.nlm.nih.gov/sra/?term=PRJNA270105 | Publicly available at NCBI Short Reads Archive (PRJNA270105). |

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
