## [Decision Letter]

Thank you for sending your work entitled “Transposable elements crowd-source mutations to rapidly fine-tune regulatory networks” for consideration at *eLife*. Your article has been favorably evaluated by Chris Ponting (Senior editor) and 3 reviewers, one of whom is a member of our Board of Reviewing Editors. One reviewer, Hideki Innan, has agreed to reveal his identity.

The Reviewing editor and the other reviewers discussed their comments before we reached this decision, and the Reviewing editor has assembled the following comments to help you prepare a revised submission.

There was general agreement that this is an exciting paper, and a substantial contribution to how our understanding of how regulatory evolution might occur. We also thought you could relatively easily do more to strengthen the case for selection having acted to increase the frequency of the TT haplotype. Simple tests based on the extent of haplotype sharing, the allele frequency distribution, etc., appear not to have been carried out. There is an obvious a priori hypothesis to test here, and multiple loci to test it with.

There was also general agreement that your title, albeit catchy, is not professional enough. Perhaps use this phrase in the discussion instead, and use a more descriptive title here. How about something like “Non-allelic gene conversion between repetitive sequences enable rapid evolutionary change at multiple regulatory sites encoded by transposable elements”?

*Reviewer #2 minor comments*:

1) It is a bit unclear on the origin of the TT holpotype. According to the description in the beginning of the subsection “MRE variation among ISX insertions in *D. miranda* strain MSH22”, it seems that the TT haplotype already existed in the ISY family, and so did the GT and TA haplotypes? But because they were rare, the authors assumed that the TT haplotype is derived? The very origin of ISX was GA, and then TT was transferred from ISX to ISY? Do I understand correctly? More detailed description would be nice.

2) I found an interesting (but not very emphasized) observation that the two sites are in strong LD in ISX, but not in ISY. A small sub-table or something in Figure 1 might help to convince readers.

3) I like that the authors looked at polymorphism. What does the frequency spectrum of TT (using all sites shown in Figure 4) look like? Is it what expected under a directional selection model? Are there any sites that are not fixed in the population yet, but there is already GA/TT polymorphism?

4) I wonder how the spatial distribution of shared sites across the TE region looks. Does your hypothesis predict an excess of shared sites around the two sites? Maybe so, but I'm not sure. Once the TT haplotypes was introduced in the ISX family, TT haplotypes may be preferentially transferred within the neo-X. In such a case, gene conversion does not matter… I don't know any theory that handles such a complicated case.

*Reviewer #3 minor comments*:

*In the beginning of the subsection “Non-allelic gene conversion is spreading the TT haplotype among ISX insertions”*: Actually the decreased linked variation in the TT haplotype is more likely to be the result of non-allelic gene conversion, rather than selection. Gene conversion may happen across hundreds nucleotides, and this in turn can be responsible of the observed decreased variation, on the scale of course of a hundreds bp around the TT sites.

Figure 1: The top-to-bottom orientation of the sequence in this figure to me is kind of counter intuitive, and is different from the more usual left-to-right orientation in Figure 3. I think Figure 1 might work better if rotated 90° counterclockwise; the authors can try and see if they also like it better.

---

## [Author Response]

*There was general agreement that this is an exciting paper, and a substantial contribution to how our understanding of how regulatory evolution might occur. We also thought you could relatively easily do more to strengthen the case for selection having acted to increase the frequency of the TT haplotype. Simple tests based on the extent of haplotype sharing, the allele frequency distribution, etc., appear not to have been carried out. There is an obvious a priori hypothesis to test here, and multiple loci to test it with*.

*There was also general agreement that your title, albeit catchy, is not professional enough. Perhaps use this phrase in the discussion instead (to encourage popular science writers to pick it up…)*, *and use a more descriptive title here. How about something like “Non-allelic gene conversion between repetitive sequences enable rapid evolutionary change at multiple regulatory sites encoded by transposable elements”?*

We have strengthened the case for selection having acted to increase the frequency of the TT haplotype, and in addition to haplotype diversity include two additional tests of directional selection using the allele frequency spectrum and nucleotide diversity. Consistent with models of directional selection, the frequency spectrum of TT-carrying haplotypes shows an excess of high frequency derived alleles, compared to the GA-carrying ISX elements, and nucleotide diversity is reduced at ISX elements where the TT haplotype is fixed, relative to GA-containing insertion sites. The results of the three statistical tests of selection are now presented in the new Figure 5. We have also modified our title as suggested.

Reviewer #2 minor comments:

*1) It is a bit unclear on the origin of the TT holpotype. According to the description in the beginning of the subsection “MRE variation among ISX insertions in* D. miranda *strain MSH22”, it seems that the TT haplotype already existed in the ISY family, and so did the GT and TA haplotypes? But because they were rare, the authors assumed that the TT haplotype is derived? The very origin of ISX was GA, and then TT was transferred from ISX to ISY? Do I understand correctly? More detailed description would be nice.*

There are multiple possibilities of how the TT haplotype could have originated—either by a double mutation in a GA-containing ISX element, or the two mutations might have occurred subsequently (and the population went through an adaptive valley), or it might have originated in an ISY element, and gene converted onto ISX. We cannot distinguish between these possibilities, but add some discussion on its origin in the Discussion section of the manuscript.

*2) I found an interesting (but not very emphasized) observation that the two sites are in strong LD in ISX, but not in ISY. A small sub-table or something in*
Figure 1
*might help to convince readers*.

We added a second panel to Figure 1 (Figure 1) showing the frequencies of all observed haplotypes for ISY and ISX elements and modified the figure legend to emphasize the strong LD of the TT haplotype on the ISX background, and also in the Discussion section of the paper.

*3) I like that the authors looked at polymorphism. What does the frequency spectrum of TT (using all sites shown in*
Figure 4*) look like? Is it what expected under a directional selection model? Are there any sites that are not fixed in the population yet*, *but there is already GA/TT polymorphism?*

We have added a supplementary figure (Figure 4–figure supplement 3) that compares the frequency spectrum across TT-carrying ISX sequences versus GA-carrying ISX sequences. Consistent with directional selection models of incomplete hitchhiking due to recombination and/or gene conversion, the frequency spectrum of TT-carrying haplotypes shows an excess of high frequency derived alleles, compared to the GA-carrying ISX elements.

*4) I wonder how the spatial distribution of shared sites across the TE region looks. Does your hypothesis predict an excess of shared sites around the two sites? Maybe so, but I'm not sure. Once the TT haplotypes was introduced in the ISX family, TT haplotypes may be preferentially transferred within the neo-X. In such a case, gene conversion does not matter… I don't know any theory that handles such a complicated case*.

The spatial distribution of shared polymorphism across the ISX element is shown on the bottom right of Figure 3. We are also not aware of any theory that would handle gene conversion and selection in multigene families, and make predictions about patterns of polymorphism at linked sites.

Reviewer #3 minor comments:

*In the beginning of the subsection “Non-allelic gene conversion is spreading the TT haplotype among ISX insertions”: Actually the decreased linked variation in the TT haplotype is more likely to be the result of non-allelic gene conversion, rather than selection. Gene conversion may happen across hundreds nucleotides, and this in turn can be responsible of the observed decreased variation, on the scale of course of a hundreds bp around the TT sites*.

We don't think this is the case. Non-allelic gene conversion can certainly reduce variation among insertions. However, while non-allelic gene conversion may decrease variation within the TT haplotype, it should equally decrease variation within the GA haplotype (assuming that they are subject to similar levels of gene conversion). The test of selection we employed shows that given its frequency, the TT haplotype harbors significantly less variation than the GA haplotype (a test that is similar in spirit to the haplotype test of selection introduced by Hudson et al, Genetics 1994). We now also show that the frequency spectrum of TT-carrying haplotypes shows an excess of high frequency derived alleles, compared to the GA-carrying ISX elements (Figure 4–figure supplement 3), which further strengthens the case that selection is operating on the TT haplotype.

Figure 1*: The top-to-bottom orientation of the sequence in this figure to me is kind of counter intuitive, and is different from the more usual left-to-right orientation in*
Figure 3*. I think*
Figure 1
*might work better if rotated 90° counterclockwise; the authors can try and see if they also like it better*.

Figure 1 is oriented the same way as Figure 3, i.e. the nucleotides of individual ISX elements are aligned from left-to-right, and the 77 different ISX insertions within the MSH22 individual are aligned top-to-bottom. We rephrase our figure legend to state this more clearly, and also modify Figure 1 and include the position of the individual ISX elements in the reference MSH22 strain, to make this figure more intuitive (as done in Figure 3).